**Data Availability Statement:** All relevant data are within the paper and its Supporting Information files.

# Qualitative insights into reasons for missed opportunities for vaccination in Kenyan health facilities

Anyie J. Li [1,2]*, Collins Tabu[3], Stephanie Shendale[4], Peter O. Okoth[5], Kibet Sergon[6], Ephantus Maree[3], Isaac K. Mugoya[7], Zorodzai Machekanyanga[8], Iheoma U. Onuekwusi[6], Ikechukwu Udo Ogbuanu[2,4]

**1** ASPPH/CDC Allan Rosenfield Global Health Fellowship and PHI/CDC Global Health Fellowship, Atlanta, GA, United States of America, **2** Global Immunization Division, Centers for Disease Control and Prevention, Atlanta, GA, United States of America, **3** National Vaccines and Immunization Program, Ministry of Health Kenya, Nairobi, Kenya, **4** Department of Immunization, Vaccines and Biologicals, World Health Organization, Headquarters, Geneva, Switzerland, **5** UNICEF Kenya Country Office, Nairobi, Kenya, **6** World Health Organization Kenya Country Office, Nairobi, Kenya, **7** Maternal and Child Survival Program, Nairobi, Kenya, **8** Inter-Country Support Team (IST)–East and Southern Africa, World Health Organization, Harare, Zimbabwe

* anyieli@cdc.gov

## Abstract

### Background

In 2016, Kenya conducted a study of missed opportunities for vaccination (MOV)—when eligible children have contact with the health system but are not fully vaccinated—to explore some of the reasons for persistent low vaccination coverage. This paper details the qualitative findings from that assessment.

### Methods

Using the World Health Organization MOV methodology, teams conducted focus group discussions among caregivers and health workers and in-depth interviews of key informants in 10 counties in Kenya. Caregivers of children <24 months of age visiting the selected health facilities on the day of the assessment were requested to participate in focus group discussions. Health workers were purposively sampled to capture a broad range of perspectives. Key informants were selected based on their perceived insight on immunization services at the county, sub-county, or health facility level.

### Results

Six focus group discussions with caregivers, eight focus group discussions with health workers, and 35 in-depth interviews with key informants were completed. In general, caregivers had positive attitudes toward healthcare and vaccination services, but expressed a desire for increased education surrounding vaccination. In order to standardize vaccination checks at all health facility visits, health workers and key informants emphasized the need for additional trainings for all staff members on immunization. Health workers and key informants

**Funding:** AJL is supported by Cooperative Agreement Number U36OE000002 from the Centers for Disease Control and Prevention (CDC) and the Association of Schools and Programs of Public Health (ASPPH) (https://www.aspph.org/study/fellowships-and-internships/) and NU2GGH002093-01-00 from the CDC and the Public Health Institute (PHI) (https://phi-cdcfellows.org/). The funders had no role in study design, data collection and analysis, decision to publish, or preparation of the manuscript.

**Competing interests:** The authors have declared that no competing interests exist.

also highlighted the negative impact of significant understaffing in health facilities, and the persistent challenge of stock-outs of vaccines and vaccination-related supplies.

## Conclusions

Identified factors that could contribute to MOV include a lack of knowledge surrounding vaccination among caregivers and health workers, inadequate number of health workers, and stock-outs of vaccines or vaccination-related materials. In addition, vaccination checks outside of vaccination visits lacked consistency, leading to MOV in non-vaccinating departments. Qualitative assessments could provide a starting point for understanding and developing interventions to address MOV in other countries.

## Background

Globally, of the 140 million children born each year, approximately 120 million receive the third dose of the diphtheria-tetanus-pertussis vaccine (DTP) [1]. This represents significant success for the Expanded Programme on Immunization (EPI), which was launched in 1974 [2–4]. Unfortunately, the 20 million children who remain un- or under-vaccinated annually reside mostly in the African region, especially in low- and middle-income countries [1]. In Kenya, although official estimates of the coverage for the third dose of DTP was 81% in 2018, 35% of the annual birth cohort still remains un- or under-vaccinated [5–7]. Some of the un- or under-vaccination of children may be attributable to Missed Opportunities for Vaccination (MOV).

A MOV includes any contact with health services by a child (or adult) who is eligible for vaccination (unvaccinated or partially vaccinated/not up to date and free of contraindications to vaccination) that does not result in the individual receiving all the vaccine doses for which he or she is eligible [8, 9]. MOV may be caused by a variety of reasons including a lack of screening for vaccination eligibility, perceived contraindications, vaccine shortages, or vaccine hesitancy [10]. MOV may be preventing countries from reaching their immunization targets. By ensuring that children who are already receiving health services are screened and vaccinated during regular health service encounters, addressing the underlying causes of MOV can significantly increase coverage and timeliness with minimal and sustainable cost [11–14].

Previous authors have documented some findings on MOV in Kenya. A review of Demographic and Health Survey data found an MOV prevalence of 42% in 2014 [15]. In 2016, another study of children of Maasai nomadic pastoralists in Kenya found a 30% prevalence of MOV [16]. Another study among children in an urban poor settlement of Nairobi, Kenya found that 22% of children who were fully immunized by 12 months had received their vaccine doses out of sequence. This suggests the occurrence of MOV, as children had contact with health services to receive some, but not all vaccines they were eligible for [17]. As seen from these reports, previous studies assessing MOV or factors related to MOV have been limited in scope and have used varying methodologies, leading to limitations in comparability and varying interpretations [15, 16, 18–23].

The World Health Organization (WHO) published a standardized methodology to assess MOV in 2015. Derived and updated from previous MOV methodologies and other studies, it places more emphasis on interventions to reduce MOV through a bottom-up approach to problem-solving [9, 10, 24, 25]. To explore the underlying reasons for persistent low vaccination coverage and investigate potential interventions to improve coverage and equity, the Kenyan National Vaccines and Immunization Program (NVIP), in collaboration with partners, conducted a study of missed opportunities for vaccination (MOV) in November 2016. This

paper details the qualitative findings from that assessment. The 2016 assessment was nested within a larger Kenya MOV assessment, which also included a quantitative component and brainstorming sessions for interventions to reduce MOV [9, 10, 24].

## Methods

The MOV assessment conducted in Kenya was based on the WHO methodology for assessment of MOV, which includes both quantitative and qualitative methods, and a desk review, to provide a comprehensive understanding of MOV [10]. The assessment aims to explore root causes and feasible solutions to address gaps in access to and utilization of vaccination services, in particular, and health services in general [9].

### Study design

This qualitative study included focus group discussions (FGDs) with caregivers and health workers and in-depth interviews (IDIs) with key informants [9, 10]. The qualitative component of the assessment is specifically designed to understand why opportunities for vaccination are being missed and what can be adjusted or done differently to mitigate MOV, in the context of Kenya. However, interventions and recommendations are not drawn from only the qualitative data, but the triangulation of other data sources.

As part of the larger Kenya MOV assessment, the in-country assessment began with a training of field team members, followed by deployment to the field for four days to sites across the country to collect data. Multi-partner brainstorming sessions followed data collection to synthesize all the data (desk review and quantitative and qualitative data) and develop an intervention plan based on the data. The assessment process concluded with a stakeholder meeting at the central government level that aimed to build consensus and advocacy for implementing an endorsed and funded intervention plan to reduce MOV. Only the results of the qualitative component of the assessment are presented in this paper.

### Field team recruitment and training

Field team members were staff from the Kenyan Ministry of Health (MoH) and various in-country immunization partners. Each field team consisted of two or three members who were trained in both quantitative and qualitative data collection over the 3 days preceding field deployment, and included at least one member with prior qualitative data collection experience who moderated the FGDs and conducted the IDIs while the other field team members took notes.

The qualitative team, comprised of one lead and two team members, had combined experience of all aspects of a qualitative study including study design, data collection, and analysis. The qualitative team led the qualitative training of the field teams. Training topics included the basics of qualitative research methods (with an emphasis on understanding the overarching research question), the purpose of conducting FGDs and IDIs for the MOV assessment, and notetaking during qualitative data collection. The qualitative team also discussed reflexivity to ensure continuous reflection by the qualitative data collectors on their role in data collection. Following in-class role playing, the MOV field teams reviewed and revised the FGD and IDI guides.

### Data collection instruments

The qualitative data collection instruments included semi-structured FGD guides for the caregiver and health worker discussions and IDI guides for key informant interviews. The guides began with easy to answer opening questions related to the topic followed by key questions.

The key questions were grouped into three sections: health and vaccination services in the community, attitudes toward vaccination and vaccination compliance, and MOV including reasons behind MOV and suggestions for reducing MOV. The guides ended with closing questions to summarize the discussion or interview. These guides were adapted for the Kenyan context from the generic guides provided in the WHO MOV methodology [10]. They were pre-tested for country-context and ease of understanding during the field staff training [9, 10]. All guides were available in English.

## Participant selection

The MOV strategy team purposively selected 10 counties to represent various geographic regions and vaccination performance levels (based on coverage of the third dose of DTP-hepatitis B-*Haemophilus influenzae* type b or *pentavalent vaccine*) [24]. Ten counties were selected as this was the maximum number logistically feasible for this assessment given time and budget constraints, but also was thought to provide enough diversity to obtain an understanding of the situation of MOV in Kenya. The counties selected were Bungoma, Kajiado, Kiambu, Kitui, Migori, Mombasa, Nakuru, Taita Taveta, Trans Nzoia, and West Pokot. Within these counties, each MOV field team (one per county) selected one or two health facilities for *qualitative* data collection. Study participants for the FGDs were selected from similar health facilities as in the quantitative arm, based on size and location; facilities that were sampled for the quantitative arm were *excluded* from the qualitative arm to avoid contamination. For their respective FGDs, caregivers and health workers were selected from either the same or different health facilities. Because the days available for data collection were limited, ease of logistical access was considered in the final selection of health facilities.

For the caregiver FGDs, MOV field teams approached caregivers with children aged <24 months at the selected health facilities and asked if they were willing to participate in the qualitative arm. Caregivers were asked for their child's age to assess eligibility. To encompass a wide range of experiences and reduce logistical constraints to participation, MOV field teams did not screen caregivers by reason for visit or if their child had a MOV. To ensure that caregivers were able to express their true opinions, the FGDs were conducted in quiet, comfortable locations beyond earshot of the health workers and other non-participants.

For the health worker FGDs, we purposively sampled health workers from diverse departments of the selected health facilities in order to capture a broad-range of perspectives. As MOV can occur at various types of visits and contact with a health facility, we ensured the inclusion of health workers who are not routinely involved in delivering vaccination services.

Key informants for IDIs were identified based on their perceived influence or insights on immunization services at different levels of the health system. Key informants included those responsible for the NVIP at the county and sub-county levels, as well as health facility directors, office managers, and administrators. Health facility-level participants for IDIs were selected from the same health facilities that were sampled for quantitative or qualitative data collection.

For the qualitative component of the MOV assessment, the overall goal was for 10 field teams to complete 10 FGDs with caregivers, 10 FGDs with health workers, and 40 IDIs with key informants. Based on experiences from previous MOV assessments in other countries, saturation was expected to occur with fewer FGDs and IDIs. However, because the 10 teams simultaneously conducted FGDs and IDIs across the 10 counties, it was impossible to review the data in real time. Each of the 10 field teams aimed to complete one FGD with caregivers, one FGD with health workers, and four IDIs (two with NVIP managers at the county or sub-county level and two with health-facility–level staff members). Qualitative data collection was complete for the team when they had achieved this goal. Individuals that participated in an

IDI were excluded from FGD participation, and vice versa. In general, health worker FGD participants had direct roles in patient care while IDI participants had management, supervisory or administrative roles.

## Data collection

Fieldwork was conducted over four days in November 2016 (November 4–5 and November 7–8). All FGD caregiver groups were same-sex groups, and field teams prioritized assigning moderators of the same sex. Moderators or interviewers began the discussion or interview by introducing themselves and the notetaker, describing the purpose of the FGD or IDI, establishing rapport with the participants or interviewees and explaining how the information would be used; in addition, they explained issues related to confidentiality and obtained informed consent from the participants. Moderators or interviewers then posed open-ended questions from the guide, starting with simple opening questions; they used probing techniques to elicit depth as needed. Prior to the FGD or IDI, no relationship was established between the moderator, interviewer, or notetaker and the participants or interviewee.

Notetakers captured key issues raised and emotional non-verbal aspects such as facial expressions and gestures in summary notes. Additionally, notetakers recorded verbatim quotes. Notes from the qualitative data collection sessions were typed up, labeled, organized, and stored on a secure website each evening. Because of privacy concerns, no audio recordings were made, and no personal information, such as name, job title, or position (for health workers and key informants), was recorded in the notes.

The FGD and IDI sessions lasted an average of 45 and 30 minutes, respectively. The sessions were conducted in the language the participants were most comfortable with. Interviewers and moderators obtained verbal consent from all participants prior to conducting FGDs or IDIs.

## Data analysis

Inductive thematic data analysis was an iterative process with re-readings of the text to discern patterns and identify major and recurring themes derived from the data. Field team members consulted as a team following each FGD or IDI to compare notes and verify verbatim quotes before sending the notes to the qualitative lead. Immediately following fieldwork, the qualitative lead conducted preliminary analyses for the immediate post-fieldwork debriefing. As a part of this process, the qualitative lead discussed major themes with all field team members, explored contradictions between sites, and further investigated unexpected findings. These discussions provided the basis for an initial set of analytic codes. Following the assessment period, the qualitative team continued refining the preliminary list of codes, rooted in the data, and used a system of manual open coding organized using Microsoft Word tables. These codes were then grouped into emerging themes through an iterative and flexible process. The final themes were determined by consensus of the qualitative team.

## Ethical approval

The Kenya MoH assessed the MOV assessment protocol and classified it as a program assessment. As such, it was exempt from further review by the Institutional Review Board. The study team obtained verbal consent from all participants prior to the FGD or IDI. The consent was recorded by the facilitator or interviewer on the guide. The verbal consent procedure was approved by the Ministry of Health Kenya.

## Results

### Participant characteristics and key themes

The MOV field teams conducted 6 FGDs with caregivers, 8 FGDs with health workers, and 35 IDIs with key informants across the 10 counties that were included in the MOV assessment (Table 1). Approximately 50 caregivers and 55 health workers participated in the FGDs; each FGD comprised of 5–13 participants. Thirty-five key informants participated in IDIs. Due to time constraints, not all teams were able to meet their qualitative data collection goal. All the caregiver FGDs were exclusively with female caregivers. Health workers who participated in the health worker FGDs were from various backgrounds and departments at the health facility. Similarly, key informants ranged from those at the county level to those at the health facility level. There were no refusals and no participants dropped out during the discussions/interviews.

The key themes that emerged from the thematic analysis included perceptions of healthcare services and vaccination; vaccination checks and integration with other services; health worker staffing shortages; stock-outs of vaccines and vaccination-related materials; and health education. The findings from the IDIs and the caregiver and health worker FGDs are summarized by key themes in the following paragraphs. Quotes included are verbatim quotes recorded during the FGDs or IDIs.

### Perceptions of healthcare services and vaccination

In general, participating caregivers expressed positive attitudes toward healthcare workers and health services. Caregivers believed that health workers were generally knowledgeable and efficient: "Nurses are good and they don't waste time" (Caregiver, Kajiado). Health workers and key informants echoed a similar sentiment with their perceptions of caregiver satisfaction: "We give our best. . . no complaints from the parents (*Others nod in agreement*)" (Health worker, Bungoma).

Although caregivers were generally satisfied with healthcare services, many participants noted that there were areas in need of improvement: "I have a feeling that clients may not be very satisfied, arising from long waiting times" (Key informant, Mombasa). Additionally, geographic inaccessibility, restricted clinic hours, and limiting the provision of certain vaccines only to specific days of the week caused some dissatisfaction among caregivers: "We come to [name] dispensary for vaccination on Wednesdays only, because that is when vaccines are available" (Caregiver, Trans Nzoia). Beyond the limited vaccination days, some also reported restricted hours even on vaccination days: "Vaccination services are given in the morning only," (Key informant, Kiambu) and "Even those that provide services daily, it is at the discretion of those in the unit to accept more clients after 12:00 pm" (Key informant, Mombasa).

Similar to their attitudes toward healthcare services, most caregivers had positive attitudes toward vaccines in general (no specific antigen was described) and to vaccination: "Immunization makes our children get better quickly from diseases that used to kill other children before" (Caregiver, Trans Nzoia). Caregivers were happy to bring their children for vaccination and reported that this was the general attitude among mothers in their communities: "Every mother brings their child for vaccination" (Caregiver, Bungoma).

Despite the generally positive attitudes, some rumors and misconceptions persisted among caregivers. Although many participating caregivers stated that they themselves did not believe the rumors, they had heard of them and knew other caregivers in their communities who did believe them. Caregivers reported feeling hesitant about multiple injections: ". . .since the introduction of a third injection [Inactivated Polio vaccine (IPV)], mothers fear that there are

**Table 1. Qualitative data collected during the Kenya missed opportunities for vaccination assessment, by county, November 2016.**

| County | Qualitative research method | | |
|---|---|---|---|
| | FGD caregivers (females only) | FGD health workers (mixed sex) | IDI key informants (mixed sex) |
| Bungoma | 1 | 1 | 4 |
| Kajiado | 1 | 1 | 4 |
| Kiambu | — | 1 | 4 |
| Kitui | 1 | 1 | 4 |
| Migori | 1 | — | 3 |
| Mombasa | 1 | 1 | 4 |
| Nakuru | — | — | 2 |
| Taita Taveta | — | 1 | 4 |
| Trans Nzoia | 1 | 1 | 3 |
| West Pokot | — | 1 | 3 |
| Total | 6 | 8 | 35 |

too many injections and thus do not bring the children on time. . ." (Caregiver, Bungoma). There were rumors about adverse events related to being given multiple vaccinations: "If you are given vaccines twice, for example, during the routine and during the campaigns, the children can get very sick or die" (Caregiver, Migori). There were also rumors about potential long-lasting, permanent health consequences related to vaccination and the quality of vaccines offered at health facilities: "Health facility gives bad injections—the vaccines themselves are bad, not the injection technique" (Caregiver, Kajiado).

## Vaccination checks and integration with other services

Field teams reported a lack of consistency across health facilities on when vaccination checks should be completed and whose responsibility it was. Some health workers and key informants reported checking vaccination status only at vaccination visits, while others would check at inpatient and outpatient visits but not at visits for other services (e.g. eye, dental, nutrition services): "If the child is not here for immunization, they don't ask, but if the child is here for immunization, they normally ask" (Key informant, Kitui). Some attributed this to the lack of integration of services within a health center, with certain departments siloed: "[Vaccination]. . .that is the work of the MCH [Maternal and Child Health] staff" (Key informant, Taita Taveta). At other facilities, checking of vaccination status was attributed to personal health worker initiative: "When I'm attending to the children in the OPD [out-patient department], it has always been my culture to check on the MCH booklet and I usually look at their immunization record. . ." (Key informant, Kitui). Overall, health workers and key informants acknowledged that health staff need to provide a holistic healthcare approach: "When telling clients, don't just brush off [and] do things shallowly; I think you need to dig deep in terms of owning that patient and finding out everything about that patient, not just why they came to the facility" (Key informant, Kitui).

## Health worker staffing shortages

Health workers and key informants discussed the inadequacy of staff resources at health facilities: "Staffing is a constant struggle" (Key informant, Kajiado). This sentiment was also echoed by participating frontline health workers: "Understaffing is really, really, really biting into services" (Health worker, Kitui). Participants explained that the lack of staff resources was not

due to a lack of trained staff in Kenya, but rather either due to a lack of resources to hire staff or due to the politics behind staff distribution among health facilities within a county.

Health workers explained how the limited staff resources affected health workers' ability to provide high quality services to patients: "Quality service is compromised because we don't have enough health workers in our health facilities, and the sufferer is a very innocent person" (Key informant, Trans Nzoia). When health workers are overwhelmed, they are unable to take the necessary time with each patient and are only able to focus on the presenting complaint: "In facilities with only a single staff, focus is on the sick child. When work is too much, children for vaccination are more likely to be told to 'come tomorrow because I am busy'" (Key informant, Mombasa). Caregivers were also dissatisfied with long wait times, which are exacerbated by staffing shortages: "In the public facilities, the children are many and thus [there is a] long queue with few health staff to attend to us" (Caregiver, Migori). This may even result in caregivers not visiting health facilities: "At a facility with only one staff conducting all the services, mothers find it hard to bear the long line. Some may not go at all" (Key informant, Bungoma).

## Stock-outs of vaccines and vaccination-related supplies

There were also discussions about the lack of vaccines and required vaccination-related supplies. Health workers specifically discussed the stock-outs of bacille Calmette-Guerin (BCG) vaccine: "No children have been vaccinated [with BCG] since November 2015; even some children are now walking having not received BCG" (Health worker, Bungoma). Stock-outs of both vaccines and syringes were reported: "The main challenge is stock-outs of vaccines and SoloShot [syringes] for administering BCG vaccine. These stock-outs, especially for SoloShot [syringes] for BCG, have become worse since devolution" (Key informant, Bungoma). These stock-outs contributed to wide-ranging effects, including health workers' inability to do their basic jobs: "The health workers are very willing to assist but they are handicapped by lack of other drugs and equipment" (Health worker, Trans Nzoia).

Stock-outs also affected caregivers' willingness to visit a health facility: "The facilities receive few vials of BCG vaccine with fewer doses than required. This makes the nurse at the facility to inform mothers to come for injections at a specific day in the week. . . the mothers will bring their children on that day, but I think we can have cases where some don't come back immediately" (Key informant, Trans Nzoia). Additionally, as caregivers return to their communities and report that certain supplies in the clinic were not available, other caregivers may be reluctant to return for fear of wasting their time and money or of not being able to receive the services they need for their children: "In addition, when there are stock-outs, parents spread the word in the community that there are no commodities (drugs, vaccines, syringes). . .so they don't come. . ." (Key informant, Bungoma).

Health workers also discussed shortages of other vaccine-related supplies, including mother-and-child health (MCH) booklets, which contain the child's health history, including vaccinations: "Bring back the mother-and-child booklets! We don't have these anymore" (Health worker, Kajiado). While substitute notebooks can hold the same information, they do not necessarily have the same structure or sentimental significance as the MCH booklets: "Mothers really enjoyed [the MCH booklets]. It gave [caregivers] a sense of connection and empowerment to have their child's health history, and it also provided an incentive to bring the card for every health contact" (Health worker, Kajiado).

Health workers and key informants also discussed ongoing problems with the vaccine cold chain. In some facilities, there were refrigerators with adequate storage space but "efficiency is questionable" (Health worker, Kitui). When problems with refrigerators arise, "the technician is not readily available and is expected to come from the county [office]" (Key informant,

Nakuru), causing delays and requiring health facilities to improvise in the meantime. Some facilities reported not having refrigerators and therefore needing to further limit vaccination days: ". . .three facilities schedule vaccination [vaccinate only on a few specified days of each month] as they do not have refrigerators" (Key informant, Bungoma).

Health workers and key informants discussed the various reasons for vaccine stock-outs, including delayed vaccine distribution at the county level, insufficient transport funds for supervision visits and for vaccine pick-up from the subcounty, and problems with the supplier. Ultimately however, many believed that these persistent problems were a result of the national devolution in 2013 in which 47 counties began setting up their own semi-autonomous institutions, impacting the healthcare system in the country. ". . .Immunization should have remained a national function. . .it was better organized then. . .we have more challenges now after devolution. . .counties do not have money to buy things like syringes because of budgetary constraints. . .." (Key informant, Bungoma). In addition to budgetary constraints, there are logistic challenges to vaccine distribution and supervision: "The old system [before devolution]. . .was a better system since now the health facilities pick up the vaccines and the supervision is erratic. There are instances she has taken a motorbike to go supervise the facilities, but some are so far-flung and no funds are available to take a vehicle there" (Key informant, Trans Nzoia).

## Health education

Participants in all FGDs and IDIs universally called for increased health education for both caregivers and health workers. Health workers acknowledged that caregivers were generally very desirous and receptive of health education: "They normally ask why, why, why and, then we answer them accordingly" (Key informant, Kitui). Many thought that basic health counseling at or before an appointment was effective, even for those small pockets of cultural and religious resistance and mobile populations in Kenya: "Health education on the importance of vaccination could improve the health-seeking behavior of the mothers" (Caregiver, Kajiado). Caregivers suggested that health facilities should engage the communities and involve their traditional leaders in advocating for vaccination services: "We need more discussions with the traditional leadership so they can hold meetings to inform parents on the importance of immunization and to dispel any traditional beliefs that there is no need to vaccinate" (Caregiver, Trans Nzoia).

Health workers also acknowledged that they both wanted and needed more health education and other on-the-job resources, a sentiment that was independently reinforced by key informants. Better education would help empower health workers to do their jobs better, in addition to reinforcing standard operating procedures and policies such as whether or not to open multi-dose vials: ". . .[health workers] send mothers away because [they are] reluctant to open a vial of vaccine for one child" (Key informant, Kajiado). Health workers and key informants agreed that health education should also be provided to new and non-NVIP staff members, not just to the seasoned NVIP staff. "There is a big knowledge gap and limited experience amongst healthcare workers offering vaccination services. . . That zeal and ownership of programmes I used to associate with the NVIP teams at sub-county and health facility level is not there in the new recruits" (Key informant, Taita Taveta). However, opportunities for training were reported to be limited; also, when training was available, the ability to participate was further limited because of understaffing: "For example, today with the immunization workers at the training, we are strapped" (Key informant, Kajiado). Some health workers also raised the issue of the devolution and how it has affected health education: "We are not doing much on

health education; since devolution, the county has not adapted the national level plans and activities that were working" (Key informant, Trans Nzoia).

## Discussion

This assessment of MOV in Kenya revealed that reasons for MOV included limited caregiver and health worker knowledge on immunization, understaffing, a lack of routine procedures to check vaccination status outside of vaccination visits, and stock-outs of vaccines or vaccination-related supplies.

Empowering caregivers and health workers with immunization information can have a positive impact on vaccine confidence and uptake [26–29]. More opportunities to increase education among caregivers and provide them with the tools to advocate for vaccines are needed to reduce MOV. Alternative avenues to increasing community empowerment should also be considered, such as engaging community and religious leaders.

Among health workers, poor knowledge regarding appropriate administration, vaccination schedules, age restrictions, opening multi-dose vials, and valid contraindications can ultimately result in MOV and low vaccination coverage. This study illustrates the frustration among health workers that training opportunities have been greatly reduced since the devolution and restructuring of the government. In 2013, Kenya began the process of devolution and the 47 counties began setting up their respective semi-autonomous institutions [30]. Although the national government retains some functions (e.g., issuing vaccination policies, standards, and guidance), many aspects of healthcare delivery (e.g., salaries, program implementation, and funding for health services) have been transferred to county jurisdictions [30]. Improved policy and coordination from the national level is needed, and health worker education needs to be prioritized at the county-level. Key informants also pointed out the benefit of educating all health workers, including those not directly involved in administering vaccines. Rumors about adverse events following immunization or negative health consequences, vaccine hesitancy related to multiple injections, and skepticism toward vaccine quality persist and can lead to MOV [31, 32]. Because these rumors and misconceptions about vaccination can occur at multiple points of contact in the health system, training the entire staff in a facility has the potential to alleviate the impact of such rumors. Previous studies have shown that from the caregiver perspective, health workers are the most trusted source of health information, and inadequate health worker knowledge about vaccination can contribute greatly to the under-immunization of children [33–35].

This study also shows that coordination of vaccination services with other health services offered at the same health facilities was poor and inconsistent. There were no standardized practices for vaccination checks on all children, and identification of children needing vaccination was left to the discretion of the health worker. Understaffing also resulted in turning children away or scheduling fewer vaccination days, further limiting the opportunities to vaccinate children. Other studies have shown that health workers might only offer certain vaccines at certain vaccination sessions and wait for a minimum number of children before opening a multi-dose vial [36, 37]. Children who are unable to visit health facilities on scheduled vaccination days can be automatically excluded [13, 38, 39]. It is important to ensure that staff outside of the NVIP program are properly sensitized about the childhood immunization schedule and are able to screen and refer eligible children for vaccination.

All groups discussed the stock-outs of both vaccines and vaccination-related supplies as a cause of MOV at their health facilities. Stock-outs of the MCH booklets made it difficult for health workers to easily assess the child's accurate age and the vaccines a child has received in the past, and can lead to confusion about which vaccine doses children are eligible to receive

[40–42]. Stock-outs of vaccines also have the potential to deter caregivers from visiting specific health facilities, and health services in general, if they had previously visited the clinic but were unable to be vaccinated because of limited vaccine supplies.

## Follow-up actions to reduce MOV and improve coverage and equity

Following field work, all field teams reconvened at the national capital to brainstorm about interventions and to create an intervention plan based on preliminary results from both the quantitative and qualitative data. The multi-partner technical working group on immunization in Kenya has endorsed this plan which aims to address the main identified causes of MOV. To improve health worker knowledge on vaccination across departments, NVIP plans to increase supportive supervision and to create an orientation package specifically targeting non-NVIP staff. Using adult learning strategies, training modules will address vaccination practices and interpersonal communication skills. Improving health worker knowledge, attitudes, and practices across all departments is expected to reduce one of the barriers to timely vaccination of eligible children [43]. The intervention plan also includes an activity to ensure an adequate supply of vaccines and vaccination-related materials by expediting the implementation of a new stock-management module across all counties. Because printing recording tools is the statutory function of counties, NVIP also prioritized efforts to provide counties with electronic copies of the latest versions of all documentation tools (e.g., monitoring charts, summary sheets, tally sheets, MCH booklets) to enable printing of copies. Finally, to clarify immunization-related policy and coordination roles between the national and county governments as a result of the devolution, the MoH plans to disseminate updated NVIP policies and guidelines, along with updated NVIP manuals and standard operating procedures.

## Limitations

The FGDs and IDIs were not recorded because of concern that recording would deter participants from speaking honestly and openly. To minimize the possibility that some of the meanings of the transcribed texts were not captured accurately, each FGD and IDI had one or two note takers who were trained to capture verbatim quotes, and teams had field staff with previous qualitative data collection experience, and additional training as moderators. Although note takers were expected to record additional basic information (including total participant size in the FGDs), this information was missing in most of the notes. Additionally, although the target number of participants per FGD was 6–8, some FGDs ended up being larger, which may have impacted group dynamics and discouraged participants from sharing their perspectives. Second, although most FGDs were moderated by someone of the same sex as the participants, this was limited by field staff availability. In a few instances of gender mismatch, cultural sensitivities around gender roles might have affected the ability of participants to express their thoughts openly. However, the likely impact of gender mismatch is minimal because given vaccination is not a sensitive topic in Kenya. Third, all caregivers included in FGDs had received services at a health facility; the perspectives captured are biased toward those who were seeking health services and may not accurately reflect the community's perspectives. We are confident that this bias had minimal impact on the validity of the findings because the focus of this study was on understanding reasons for MOV in healthcare settings. Nonetheless, these findings are not generalizable to the entire Kenyan community. Finally, these findings identify factors that might contribute to MOV, but these factors may not be causal.

## Conclusion

In some countries the contribution of MOV to under-vaccination of children has been assessed using existing data, such as Demographic and Health Surveys, Multiple Indicator Cluster Surveys, and other administrative health facility data [22, 44]. In situations where such secondary data exist, conducting the quantitative MOV surveys might not be necessary. However, since existing secondary surveys rarely explore the underlying reasons for MOV, countries needing further details on the reasons behind MOV may opt to conduct the *qualitative* component of the MOV assessment [9]. To minimize cost and increase efficiencies, these qualitative surveys may be integrated with other regularly scheduled program reviews such as EPI reviews and coverage surveys [45, 46]. Although a full (quantitative and qualitative) assessment was conducted in Kenya, the data generated from the qualitative component provided important data which contributed to the process of brainstorming solutions and implementing interventions to address MOV.

As of 2019, 12 countries have implemented the updated WHO methodology to reduce MOV across 4 WHO regions—African Region, Eastern Mediterranean Region, South-East Asia Region and Western Pacific Region. In addition, many more countries continue to view the MOV strategy as a potential quick-win strategy for improving vaccination coverage, and many more implementations are expected in the near future. Accumulated lessons from these countries, especially experiences with interventions to reduce MOV, will provide an additional strategy in the tool kit of immunization program managers at national and sub-national levels. Forthcoming inter-country and inter-regional comparisons will also assist global immunization partners in prioritizing interventions to make progress towards Global Vaccine Action Plan coverage and equity targets and indicators [47].

## Supporting information

**S1 File. Clarification of ethical review for standard program reviews in Kenya.**
(PDF)

**S2 File. Missed opportunities for vaccination focus group discussion guide for health workers: Kenya, 2016.**
(PDF)

**S3 File. Missed opportunities for vaccination focus group discussion guide for caregivers: Kenya, 2016.**
(PDF)

**S4 File. Missed opportunities for vaccination in-depth interview guide for key informants: Kenya, 2016.**
(PDF)

## Acknowledgments

The authors thank the caregivers, health workers, and healthcare administrators who gave of their time to participate in the focus group discussions and in-depth interviews. They also acknowledge the assistance of the entire Kenya *MOV Team*, Kenya Ministry of Health, the country offices of the World Health Organization, the Maternal and Child Survival Program (MCSP), the Clinton Health Access Initiative (CHAI), Health NGOs Network (HENNET), UNICEF, the Inter-religious council of Kenya, the American Red Cross, and other local immunization partners during the assessment and subsequent implementation of interventions to reduce missed opportunities for vaccination in Kenya.

### Disclaimer

The authors alone are responsible for the views expressed in this article, which do not necessarily represent the views, decisions, or policies of the institutions with which the authors are affiliated.

## Author Contributions

**Conceptualization:** Collins Tabu, Kibet Sergon, Ikechukwu Udo Ogbuanu.

**Data curation:** Anyie J. Li, Stephanie Shendale, Ikechukwu Udo Ogbuanu.

**Formal analysis:** Anyie J. Li, Stephanie Shendale, Ikechukwu Udo Ogbuanu.

**Investigation:** Anyie J. Li, Collins Tabu, Stephanie Shendale, Peter O. Okoth, Kibet Sergon, Ephantus Maree, Isaac K. Mugoya, Zorodzai Machekanyanga, Iheoma U. Onuekwusi, Ikechukwu Udo Ogbuanu.

**Methodology:** Collins Tabu, Kibet Sergon, Iheoma U. Onuekwusi, Ikechukwu Udo Ogbuanu.

**Project administration:** Collins Tabu, Kibet Sergon, Ephantus Maree, Ikechukwu Udo Ogbuanu.

**Resources:** Collins Tabu, Stephanie Shendale, Peter O. Okoth, Kibet Sergon, Ephantus Maree, Iheoma U. Onuekwusi, Ikechukwu Udo Ogbuanu.

**Supervision:** Anyie J. Li, Collins Tabu, Stephanie Shendale, Peter O. Okoth, Kibet Sergon, Isaac K. Mugoya, Zorodzai Machekanyanga, Iheoma U. Onuekwusi, Ikechukwu Udo Ogbuanu.

**Validation:** Anyie J. Li, Stephanie Shendale, Ikechukwu Udo Ogbuanu.

**Writing – original draft:** Anyie J. Li.

**Writing – review & editing:** Collins Tabu, Stephanie Shendale, Peter O. Okoth, Kibet Sergon, Isaac K. Mugoya, Zorodzai Machekanyanga, Iheoma U. Onuekwusi, Ikechukwu Udo Ogbuanu.

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
