## [Decision Letter · Decision Letter 0]

14 Jan 2020

PONE-D-19-31528

Qualitative Insights into Reasons for Missed Opportunities for Vaccination in Kenyan Health Facilities

PLOS ONE

Dear Ms. Li,

Thank you for submitting your manuscript to PLOS ONE. After careful consideration, we feel that it has merit but does not fully meet PLOS ONE’s publication criteria as it currently stands. Therefore, we invite you to submit a revised version of the manuscript that addresses the points raised during the review process.

We would appreciate receiving your revised manuscript by Feb 28 2020 11:59PM. To enhance the reproducibility of your results, we recommend that if applicable you deposit your laboratory protocols in protocols.io, where a protocol can be assigned its own identifier (DOI) such that it can be cited independently in the future. For instructions see: http://journals.plos.org/plosone/s/submission-guidelines#loc-laboratory-protocols

We look forward to receiving your revised manuscript.

Kind regards,

Holly Seale

Academic Editor

PLOS ONE

Journal Requirements:

2. Please specify in your ethics statement: a) whether the ethics committee approved the verbal/oral consent procedure, b) why written consent could not be obtained, and c) how verbal/oral consent was recorded.

3. Please include in your Methods section the date ranges over which you recruited participants to this study.

4. Please include additional information regarding the survey or questionnaire used in the study and ensure that you have provided sufficient details that others could replicate the analyses. If you developed and/or translated a questionnaire as part of this study and it is not under a copyright more restrictive than CC-BY, please include a copy, in both the original language and English, as Supporting Information.

We note that one or more of the authors are employed by a commercial company: John Snow, Inc. Country Office, Nairobi, Kenya.

6. We note that Figure #1 in your submission contain [map/satellite] images which may be copyrighted. All PLOS content is published under the Creative Commons Attribution License (CC BY 4.0), which means that the manuscript, images, and Supporting Information files will be freely available online, and any third party is permitted to access, download, copy, distribute, and use these materials in any way, even commercially, with proper attribution. For these reasons, we cannot publish previously copyrighted maps or satellite images created using proprietary data, such as Google software (Google Maps, Street View, and Earth). For more information, see our copyright guidelines: http://journals.plos.org/plosone/s/licenses-and-copyright.

a.    You may seek permission from the original copyright holder of Figure #1 to publish the content specifically under the CC BY 4.0 license. 

Reviewers' comments:

Reviewer's Responses to Questions

**Comments to the Author**

1. Is the manuscript technically sound, and do the data support the conclusions?

Reviewer #1: Yes

Reviewer #2: Yes

2. Has the statistical analysis been performed appropriately and rigorously? 

Reviewer #1: N/A

Reviewer #2: N/A

3. Have the authors made all data underlying the findings in their manuscript fully available?

Reviewer #1: No

Reviewer #2: Yes

4. Is the manuscript presented in an intelligible fashion and written in standard English?

Reviewer #1: Yes

Reviewer #2: Yes

5. Review Comments to the Author

Reviewer #1: Background

1. Line 45: Include more information about immunization coverage in Kenya and the factors that have been identified to predict low immunization coverage in the country.

2. Line 47 – 50: To improve flow, you may consider moving this part to the last paragraph of the background section.

Methodology

1. The term “cross-sectional” is commonly associated with quantitative studies. Given that the philosophical assumptions of this class of studies is markedly different from qualitative, I’d suggest that it should be avoided here.

2. The rationale for using grounded theory in this manuscript is debatable. MOV is already a pre-existing concept. For this reason, the goal of the study was not to develop new theories (and a theoretical framework) about MOV. In my opinion, this study leans more towards a critical realism because of the causal-explanatory approach that was adopted to describe factors that are responsible for MOV. Moreover, the study builds on already existing knowledge by exploring Kenya’s context and identified factors that are relatively similar to those identified in existing literature. Overall, there is a need to extensively and convincingly justify the methodological paradigm that is being adopted in the study design sub-section.

3. Line 80: Include a sub-section on reflexivity.

4. Line 100: Please remove “sample size”

5. Line 101: What were the criteria for this purposive selection? How do the counties differ from each other? Why 10?

6. What were the criteria used to stop data collection?

Reviewer #2: This paper presents a qualitative study which examined the reasons for missed opportunities for vaccinations in Kenyan Health facilities. I do believe this paper would be of interest to readers of PLOS ONE however, the following issues outlined below need to be addressed.

Introduction:

- Lines 47-50: can this be moved to the final paragraph of the introduction?

- I think more references to other studies conducted in Kenya on vaccine uptake needs to be added in the introduction. That way it paints a clearer picture of the the research gap as well as how this study fits into the literature. I don't quite follow at the moment and it is not quite clear.

Methods:

- Can you clarify what type of thematic analysis was used ? It seems like an inductive approach was used but this needs to be mentioned in the methods so that it is clear to the reader.

Discussion:

- this is currently very long and needs to be streamlined. A lot of time is spent regurgitating the results and little time is spent engaging with the literature.

- Did you look for any differences and similarities within your sample? Perhaps also look at other studies that have examined vaccine uptake in Kenyan populations globally? And see if some of the issues identified are the same or different? Obviously there would be a huge distinction between those in low income vs. high income countries but this might add some interesting nuances.

6. PLOS authors have the option to publish the peer review history of their article (what does this mean?). If published, this will include your full peer review and any attached files.

Reviewer #1: Yes: Charles Shey Wiysonge

Reviewer #2: No

---

## [Author Response · Author response to Decision Letter 0]

6 Feb 2020

Thank you for your review. Please see the reviewer response table and the manuscript with track changes for detailed information on the changes made.

---

## [Decision Letter · Decision Letter 1]

10 Mar 2020

Qualitative Insights into Reasons for Missed Opportunities for Vaccination in Kenyan Health Facilities

PONE-D-19-31528R1

Dear Dr. Li,

We are pleased to inform you that your manuscript has been judged scientifically suitable for publication and will be formally accepted for publication once it complies with all outstanding technical requirements.

With kind regards,

Holly Seale

Academic Editor

PLOS ONE

Additional Editor Comments (optional):

Reviewers' comments:

Reviewer's Responses to Questions

**Comments to the Author**

1. If the authors have adequately addressed your comments raised in a previous round of review and you feel that this manuscript is now acceptable for publication, you may indicate that here to bypass the “Comments to the Author” section, enter your conflict of interest statement in the “Confidential to Editor” section, and submit your "Accept" recommendation.

Reviewer #1: All comments have been addressed

2. Is the manuscript technically sound, and do the data support the conclusions?

Reviewer #1: Yes

3. Has the statistical analysis been performed appropriately and rigorously? 

Reviewer #1: N/A

4. Have the authors made all data underlying the findings in their manuscript fully available?

Reviewer #1: No

5. Is the manuscript presented in an intelligible fashion and written in standard English?

Reviewer #1: Yes

6. Review Comments to the Author

Reviewer #1: (No Response)

7. PLOS authors have the option to publish the peer review history of their article (what does this mean?). If published, this will include your full peer review and any attached files.

Reviewer #1: No

---

## [Editor Report · Acceptance letter]

13 Mar 2020

PONE-D-19-31528R1 

Qualitative Insights into Reasons for Missed Opportunities for Vaccination in Kenyan Health Facilities 

Dear Dr. Li:

I am pleased to inform you that your manuscript has been deemed suitable for publication in PLOS ONE. Congratulations! Your manuscript is now with our production department. 

With kind regards,

on behalf of

Dr. Holly Seale 

Academic Editor

PLOS ONE